

# Genomic DNA extraction optimization and validation for genome sequencing using the marine gastropod Kellet's whelk

Benjamin N. Daniels[1,2], Jenna Nurge[2], Olivia Sleeper[2], Andy Lee[3], Cataixa López[4], Mark R. Christie[3,5], Robert J. Toonen[4], Crow White[1] and Jean M. Davidson[2]

[1] Center for Coastal Marine Sciences, California Polytechnic State University - San Luis Obispo, San Luis Obispo, CA, United States of America
[2] Biological Sciences Department, California Polytechnic State University - San Luis Obispo, San Luis Obispo, CA, United States of America
[3] Department of Biological Sciences, Purdue University, West Lafayette, IN, United States of America
[4] Hawai'i Institute of Marine Biology, University of Hawai'i at Mānoa, Kāne'ohe, HI, United States of America
[5] Department of Forestry and Natural Resources, Purdue University, West Lafayette, IN, United States of America

Corresponding author
Benjamin N. Daniels,
ben.daniels255@gmail.com

## ABSTRACT

Next-generation sequencing technologies, such as Nanopore MinION, Illumina Hiseq and Novaseq, and PacBio Sequel II, hold immense potential for advancing genomic research on non-model organisms, including the vast majority of marine species. However, application of these technologies to marine invertebrate species is often impeded by challenges in extracting and purifying their genomic DNA due to high polysaccharide content and other secondary metabolites. In this study, we help resolve this issue by developing and testing DNA extraction protocols for Kellet's whelk (*Kelletia kelletii*), a subtidal gastropod with ecological and commercial importance, by comparing four DNA extraction methods commonly used in marine invertebrate studies. In our comparison of extraction methods, the Salting Out protocol was the least expensive, produced the highest DNA yields, produced consistent high DNA quality, and had low toxicity. We validated the protocol using an independent set of tissue samples, then applied it to extract high-molecular-weight (HMW) DNA from over three thousand Kellet's whelk tissue samples. The protocol demonstrated scalability and, with added clean-up, suitability for RAD-seq, GT-seq, as well as whole genome sequencing using both long read (ONT MinION) and short read (Illumina NovaSeq) sequencing platforms. Our findings offer a robust and versatile DNA extraction and clean-up protocol for supporting genomic research on non-model marine organisms, to help mediate the under-representation of invertebrates in genomic studies.

## INTRODUCTION

Rapidly advancing next generation sequencing technologies such as whole genome sequencing, genotyping-in-thousands by sequencing (GT-seq), and restriction-site-associated DNA sequencing (RAD-seq), are becoming more available and affordable for non-model organisms (*Park & Kim, 2016*; *Ellegren, 2014*; *Van Wyngaarden et al., 2017*; *Bootsma et al., 2020*). However, applying these technologies to marine invertebrate species, which are nearly all non-model organisms, is hindered by challenges in extracting and purifying their genomic DNA (*Boughattas et al., 2021*). DNA extraction of marine invertebrate species can be riddled with contaminants that result in poor sequencing results (*Adema, 2021*; *Chakraborty, Saha & Neelavar Ananthram, 2020*; *Panova et al., 2016*; *Angthong et al., 2020*). The contaminants are commonly attributed to mucopolysaccharides that co-precipitate with genomic DNA, increasing the viscosity of the sample during extractions and preventing the activity of enzymes as well as proper homogenization of mixtures (*Boughattas et al., 2021*). As a result, researchers studying marine invertebrates species have had to dedicate extensive time and resources to troubleshooting and optimizing their genomic DNA extraction methods (*Ferrara et al., 2006*; *Panova et al., 2016*; *Boughattas et al., 2021*; *Adema, 2021*).

This study aims to develop and test DNA extraction protocols and generate an optimal protocol for the marine gastropod, Kellet's whelk (*Kelletia kelletii*), for supporting genomic research on the species and marine invertebrates in general. Kellet's whelk is a subtidal gastropod distributed along the North American west coast (*Zacherl, Gaines & Lonhart, 2003*), a significant predator in kelp forest ecosystems (*Halpern, Cottenie & Broitman, 2006*), and a species of interest to commercial fisheries (*Aseltine-Neilson et al., 2006*). Kellet's whelk currently supports the second-largest molluscan commercial fishery in California in both landings and ex-vessel value (*CDFW, 2023*). Furthermore, Kellet's whelk recently exhibited a $\sim$ 300 km northward range expansion (*Herrlinger, 1981*), potentially driven by intensifying El Niño Southern Oscillation (ENSO) oceanographic conditions due to climate change (*Zacherl, Gaines & Lonhart, 2003*; *Harley et al., 2006*). Current research on the species seeks to use genomic-based assignment testing (*sensu Christie et al., 2017*) to investigate this hypothesis and, more broadly, elucidate spatio-temporal patterns of gene flow and population dynamics of Kellet's whelk. Such information holds significant promise in addressing pivotal marine ecological and evolutionary inquiries and providing valuable support for the sustainable management of the Kellet's whelk fishery, as well as that of other ecologically and economically important coastal species with analogous life history traits (*e.g.*, large populations, slow growth, sedentary demersal adults, and pelagic larvae) such as spiny lobster (*Panulirus interruptus*), kelp bass (*Paralabrax clathratus*), sheephead (*Semicossyphus pulcher*), and some rockfish (*Sebastes atrovirens*) (*Allen, Pondella & Horn, 2006*; *Froese & Pauly, 2011*).

An experimental design for revealing genomic features of Kellet's whelk could be supported by applying next generation sequencing technologies to a large sample size of individuals in parent and recruit (1–2 year old Kellet's whelk individuals) populations across the species' biogeographic range and over multiple years (*Christie et al., 2017*). For instance,

the application of GT-seq and RAD-seq techniques targeting neutral loci, along with GT-seq focusing on geographically-associated loci harboring putatively adaptive alleles, can offer insights into the underlying adaptive genetic mechanisms and historical population dynamics within the species (*Reitzel et al., 2013*). Moreover, these approaches can delineate spatially distinct population units characterized by unique genotypic patterns, facilitating the assignment testing of recruits to adult populations, thereby revealing population connectivity in the species (*Malde, 2014*; *Christie et al., 2017*). To support such genomics research, a reliable method for generating high-molecular weight (HMW) genomic DNA is required from each sample. Furthermore, analysis of a large sample size (*e.g.*, 1,000s of individuals, necessary for generating a complete population connectivity matrix; *Christie et al., 2017*) requires an optimized high-throughput HMW DNA extraction methodology. To help meet this need for Kellet's whelk and potentially other non-model marine invertebrate species, we tested and compared the efficacy of DNA extraction protocols on Kellet's whelk tissue. We evaluated four DNA extraction methods from marine invertebrate studies: (1) Qiagen Blood and Tissue kit (QIAGEN, MD, USA) (*White & Toonen, 2008*; *Song, Thomas & Edwards, 2019*), (2) Zymo Quick-DNA HMW Magbead kit (Zymo, CA, USA) (*Akinde et al., 2022*), (3) Phenol Chloroform protocol (PCI)(Thermo Fisher Scientific, MA, USA) (*Nagel, Sewell & Lavery, 2015*), and, (4) an optimized version of a Salting out protocol (*Li et al., 2011*; *Ferrara et al., 2006*). For each, we conducted DNA extraction on 4–5 tissue samples, then compared the quality of DNA extractions produced, as well as the difficulty, toxicity and cost of the extraction method. We then used our test results and lessons learned from other marine genomic research projects to develop a modified protocol intended to optimally balance efficiency, cost, toxicity, and quality of extracting Kellet's whelk DNA. We validated our protocol by evaluating DNA yield, DNA quality, and DNA purity generated by DNA extractions on an independent set of 16 tissue samples. We then applied our protocol to over three thousand tissue samples. We conducted the DNA extractions in batches using a group of trained student researchers, and assessed the quality of the DNA extractions produced on a random subset of the samples. We also tested a subset of the extractions for both RAD-seq and GT-seq methods. Finally, we tested our protocol by applying DNA extractions generated from it to whole genome sequencing using both long read sequencing (ONT MinION) and short read sequencing (Illumina Novaseq).

## METHODS

Portions of this text were previously published as part of a preprint (https://www.biorxiv.org/content/10.1101/2023.07.31.551321v2). We tested four DNA extraction methods used in previous marine invertebrate studies. Each method was applied to tissue samples dissected from adult Kellet's whelk (between 60–120 mm in length) collected in the wild and maintained prior to dissection in flow-through filtered seawater aquaria at the California Polytechnic State University research pier, located in Avila Beach, California, USA. The adult whelks were collected from sub-tidal reefs located near Monterey (36.6181670N, 121.897W), Naples (34.4219670N, 119.952283W), Diablo Canyon (35.2244500N, 120.877483W), and Point Loma (32.665333N, 117.261517W) California

in 2019, transported live to the aquaria, and maintained under ambient conditions with food in the form of frozen seafood provided ad libitum (CDFW Scientific Collection Permit 8018 to C.W.). Tissue samples were dissected in March 2021. Approximately 200 mg tissue was dissected non-invasively from the foot of the individual, immediately frozen in liquid nitrogen, then transported to the Center for Applied Biotechnology at California Polytechnic State University and stored at −80 °C. A portion of each tissue sample was used for DNA extractions as specified in each method.

Five Kellet's whelk individuals were dissected for the Qiagen and Zymo kit extractions with each sample having a duplicate (10 total extractions conducted using the Qiagen kit and 10 total extractions conducted using the Zymo kit) in March 2021. Four Kellet's whelk individuals were dissected for the Salting out and PCI protocols with duplicates for the Salting out protocol but not for the PCI protocol (eight total extractions conducted using the Salting out protocol and four total extractions conducted using the PCI protocol) in May 2021.

Each method was evaluated based on protocol difficulty, cost per extraction, DNA yield, DNA quality, and toxicity. Difficulty was measured on a three-level qualitative scale. Easy: requires very little wet bench expertise - simple pipetting volumes and reagents as well as basic techniques. Medium: requires some wet bench experience—simple pipetting volumes and reagents with more advanced techniques (*e.g.*, bead binding, DNA washing and drying). Hard: requires more extensive wet bench experience—precise pipetting and handling of toxic reagents (*e.g.*, pipetting layered solutions, working in fume hoods and more extensive use of personal protective equipment (PPE)). Cost per extraction was calculated by the sum of costs per method (*e.g.*, price of Qiagen kit, cost of all reagents for PCI method) divided by the number of possible DNA extractions. DNA yield was measured by a Qubit fluorometer using the DNA Broad Range assay (Invitrogen, USA). DNA quality was evaluated based on gel electrophoresis using ∼250 ng DNA per sample (or 10 µL sample if concentration was under 25 µg/µL) on a 1% agarose gel at 80V for 45 min with a 1kb DNA ladder. Gel electrophoresis has been shown to be a robust method for quantifying DNA quality (*Gaither et al., 2011*; *Bag et al., 2016*; *Green & Sambrook, 2012*). Quality was measured as good: high molecular weight (HMW) band; medium: HMW band with degradation; or, bad: digested. Toxicity of each method was measured by the reagent classification in accordance with Occupational Safety and Health Administration(OSHA), International Agency for Research on Cancer (IARC), or National Toxicology Program (NTP). Low toxicity: no ingredients are registered as carcinogenic, aspirationally or reproductively toxic. Medium toxicity: no ingredients are registered as carcinogenic, aspirationally or reproductively toxic, but some ingredients are registered as causing skin irritation, serious irritation, or damage to eyes. High toxicity: ingredients are registered as carcinogenic, aspirationally or reproductively toxic.

## Qiagen DNeasy blood and tissue kit

DNA extraction procedures followed the manufacturer's protocol. For the Qiagen DNeasy Blood and Tissue kit, 25 mg tissue per sample was cut into small pieces and placed in a 1.5 mL microcentrifuge tube. 180 µL Buffer ATL and 20 µL Proteinase K were added.

The sample was mixed by vortexing, then incubated at 56 °C overnight. The sample was vortexed for 15 s, then 200 µL Buffer AL was added. The sample was again vortexed, then 200 µL 100% ethanol was added. The sample was vortexed one last time, then pipetted into a DNeasy Mini spin column placed in a two mL collection tube. The sample was centrifuged at 6,000× g for 1 min and the flow-through was discarded. The DNeasy Mini spin column was placed in a new two mL collection tube, 500 µL Buffer AW1 was added, and then the sample was centrifuged at 6,000× g for 1 min. The flow-through was discarded, 500 µL Buffer AW2 was added, and then the sample was centrifuged at 20,000× g for 3 min. The flow-through and collection tube were discarded and the DNeasy Mini spin column was placed in a clean 1.5 mL microcentrifuge tube. 100 µL Buffer AE was added to the DNeasy membrane and incubated at room temperature for 1 min. The sample was centrifuged at 6,000 x g for 1 min to elute the DNA. The previous step was repeated with 100 µL Buffer AE to maximize DNA yield. Eluted DNA was stored at −20 °C. Protocol can be found in the SI 7.

**Zymo Quick-DNA HMW Magbead kit**

DNA extraction procedures followed the manufacturer's protocol. For the Zymo Quick-DNA HMW Magbead kit, 25 mg tissue per sample was added to a 1.5 mL microcentrifuge tube. 95 µL DNA Elution Buffer, 95 µL Biofluid and Solid Tissue Buffer, and 10 µL Proteinase K were added to the sample. The sample mixture was pipette mixed 5 times and incubated at 55 °C for 1–3 h until tissue became soluble. The sample was centrifuged at 10,000× g for 1 min. The supernatant was removed and transferred to a new clean 1.5 mL microcentrifuge tube. 400 µL Quick-DNA MagBinding Buffer was added to the sample and mixed by pipetting. 33 µL MagBinding Beads were added to the sample, pipette mixed five times, and placed on a shaker for 10 min. The sample was placed on a magnetic stand until the beads separated from the solution. The supernatant was removed and discarded and the sample was taken off of the magnetic stand. 500 µL Quick-DNA MagBinding Buffer was added to the sample, pipette mixed five times, and placed on a shaker for 5 min. The sample was placed on a magnetic stand until the beads separated from the solution. The supernatant was removed and discarded and the sample was taken off of the magnetic stand. 500 µL DNA Pre-Wash Buffer was added to the sample and pipette mixed 10 times. The sample was placed on a magnetic stand until the beads separated from the solution. The supernatant was removed and discarded and the sample was taken off of the magnetic stand. 900 µL g-DNA Wash Buffer was added to the sample and pipette mixed 10 times. All liquid was transferred into a new microcentrifuge tube and placed on a magnetic stand until the beads separated from the solution.The supernatant was removed and discarded and the sample was taken off of the magnetic stand. The DNA wash step was repeated. The beads were dried for 20 min at room temperature and 50 µL DNA Elution Buffer was added. The sample mixture was pipette mixed 20 times and incubated at room temperature for 5 min. The sample was placed on a magnetic stand until the beads separated from the solution. The supernatant (eluted DNA) was removed and transferred to a new clean 1.5 mL microcentrifuge tube. Eluted DNA was stored at −20 °C. Protocol can be found in the SI 8.

## PCI Protocol

The PCI DNA extraction procedure followed an adopted method from Thermo Fisher Scientific. For each sample, 30 mg tissue was homogenized mechanically in 1% SDS cell lysis buffer, 0.5M EDTA, and Proteinase K. The sample was then placed in a 65° C water bath for 1 h, and vortexed every 20 min. 1× volume Phenol Chloroform (PCI) was added to the sample and vortexed. The sample was centrifuged for 5 min at 16,000× g and the upper aqueous phase was transferred to a new tube. 20 μg Glycogen, 0.5× volume Ammonium Acetate 7.5 M, and 2.5× volume 100% ethanol were added to the sample. The sample was pipette mixed and kept at −20 °C overnight, then centrifuged at 4 °C for 30 min at 16,000× g and the supernatant was removed. 150 μL 70% ethanol was added to the sample, it was centrifuged at 4 °C for 2 min at 16,000× g and the supernatant was removed. The previous step was repeated. DNA was dried in a speedvac for 2 min and resuspended in 200 μL Elution Buffer. Eluted DNA was stored at −20 °C. Protocol can be found in the SI 9.

## Salting out protocol

The Salting out protocol was adopted from *Li et al. (2011)*. For each sample, 30 mg tissue was homogenized mechanically in 1% SDS cell lysis buffer, 0.5M EDTA, and Proteinase K. The sample was placed in a 65 °C water bath for 1 h, and vortexed for 5 s every 20 min. RNase A was added, the sample was lightly vortexed, then incubated at 37 °C for 10 min. 150 μL 7.5M Ammonium Acetate was added to the sample and mixed by vortexing. The sample was incubated at 4 °C for 5 min and centrifuged at 12,000× g for 10 min at 4 °C. The supernatant was transferred to a new tube and 60 μL 7.5M Ammonium Acetate was added. The sample was mixed by vortexing, incubated at 4 °C for 5 min, and centrifuged at 12,000× g for 10 min at 4 °C. The supernatant was transferred to a new tube and 1x volume 100% isopropanol was added. The sample was inverted 50 times and centrifuged at 8,000× g for 5 min at 4 °C. The supernatant was discarded and 400 μL 70% ethanol was added. The sample was inverted 3 times and centrifuged at 4 °C at 16,000× g for 5 min. The previous two steps were repeated. The sample was dried in a speedvac at 30 °C for 4 min. DNA was resuspended in 100 μL TE buffer and incubated at 4 °C overnight. Eluted DNA was stored at −20 °C. Protocol can be found in the SI 10.

## Validation

The protocol selected as the most optimal among the four methods (based on the four selection criteria; see above and Results), was then validated on an independent set of 16 tissue samples from 9 adult and 7 recruit Kellet's whelks collected in the wild at 15 sub-tidal reef locations across the species' range in 2015, 2016, and 2017 (CDFW Scientific Collection Permit 8018 to C.W.) (Fig. 1). Tissue samples were collected from the foot of each individual Kellet's whelk using a modified scalpel. Tissue samples from adults were collected non-lethally in the field, while recruits were collected whole in the field (because of their small size) and tissue samples were collected from them in the lab. Adult tissues and whole recruits collected in the field were frozen on dry ice or in liquid nitrogen, then transported to Cal Poly and stored at −80 °C until used for this study in 2021 and 2022. DNA extractions were evaluated based on DNA yield, DNA quality, and DNA purity.

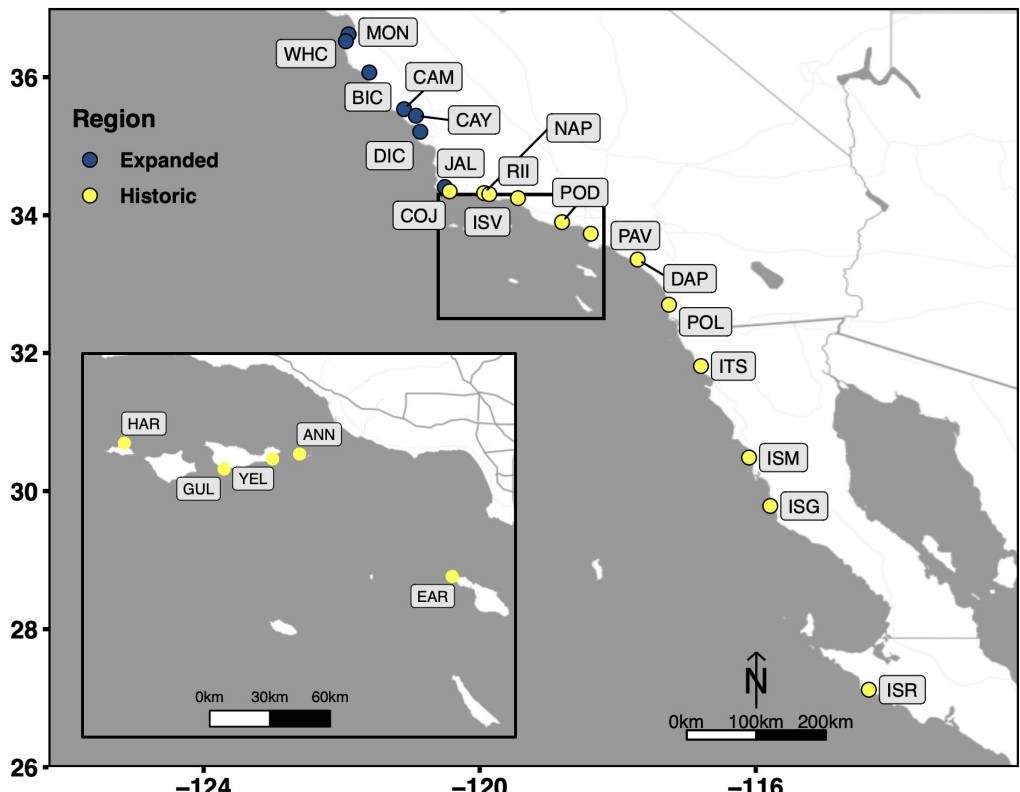

**Figure 1** **Collection sites which span the entire biogeographic range of Kellet's whelk.** Kellet's whelk populations north of Pt. Conception represents a recent range expansion into colder-water habitat (*Herrlinger, 1981*). See Supplementary Material section for location coordinates (Table S1).

DNA yield and DNA quality were assessed using the same methods described above for comparing the four DNA extraction methods. DNA purity was measured by absorbance at 260/280 (contamination of proteins absorb at ~280 nm, ideal ratio = ~1.8) and 260/230 (contamination of salts, carbohydrates, and/or phenols absorb at ~230 nm, ideal ratio = ~1.8–2.2) (*Thermo Fisher Scientific, 2009*) using a Nanodrop-1000 spectrophotometer (Invitrogen, Waltham, MA, USA).

## Scaling-up

The selected optimal method was applied to 3,325 Kellet's whelk tissue samples and assessed by DNA yield and DNA quality using the same methods above. The tissue samples represent 2,026 adults and 1,299 recruits from the same set of wild-collected samples used in the validation step described above, but covering more sites (Table S1; Fig. 1). The extraction procedure was conducted by 13 undergraduate students at California Polytechnic State University in the Center for Applied Biotechnology from January 2022 to May 2022. DNA was extracted from samples in groups of 20 each week by each student. Three extractions were randomly selected from each set of 20 extractions bi-weekly for quality check assurance throughout the extraction procedure. Quality checks were conducted based on DNA yield and DNA quality, using the same methods described above for comparing the four DNA

extraction methods. Sample quality and DNA concentration correlations were tested in R and visualized using ggplot2.

## RAD-seq and GT-seq

A subset of the scaled-up DNA extractions were tested for sequencing quality on both GT-seq and RAD-seq platforms. A range of varying locations, years, and tissue types were used for each subset (Tables S3, S4, S5). For GT-seq, 96 DNA extractions were shipped frozen to Twin Falls, ID, USA and processed at the GTseek LLC laboratory. Briefly, exonuclease 1 was added to each gDNA extraction. gDNA was amplified using the set GT-seq primers. PCR products were indexed for Illumina sequencing with i7 and i5 index. Samples were normalized using Nate's Plates Tagging and Normalization kit (SI 18). Size selection was conducted using SPRI beads. In depth methods can be found in the Supplementary Material section. Further optimization was conducted for GT-seq using the Zymo Clean and concentrator kit on aliquots of the same samples initially tested. The same methods were used for sequencing these samples. The genotyping pipeline used can be found at https://github.com/GTseq/GTseek_utils. For RAD-seq, 172 DNA extractions were shipped frozen to the Hawai'i Institute of Marine Biology, Hawai'i, USA and processed at the ToBo genomics lab (https://tobolab.org/). The gDNA digestion was performed with isoschizomer restriction enzymes (New England Bio-Labs, Ipswich, MA, USA). Libraries were generated using the KAPA Hyper Prep DNA kit (Roche Sequencing and Life Science) and then shipped frozen to the University of California, Davis, CA, USA (UCD) and sequenced at the UCD Genome Center through HiSeq2500. RAD-seq reads were filtered by their quality scores and the presence of the RADcutsite using process_radtags in Stacks (*Rivera-Colón & Catchen, 2022*). Protocols can be found in the SI 11–12.

## Library preparation

To optimize and validate the selected DNA extraction method for next generation sequencing, a library from one of the DNA extractions generated using the Salting out protocol was prepared for Nanopore MinION sequencing using the rapid sequencing kit (SQK-RAD004; ONT, Oxford, UK) without any further cleaning or repair. Three DNA libraries from the same DNA extraction were prepared with further purification and concentration using the Genomic DNA Clean and Concentrator kit (Zymo Research, Irvine, CA, USA). The DNA was then selected for high molecular weight DNA using the PacBio Short Read Eliminator kit (PacBio, Menlo Park, CA, USA). Concentration was assessed using the dsDNA BR assay on a Qubit fluorometer (Thermo Fisher Scientific, Waltham, MA, USA). Two libraries were prepared for Nanopore MinION sequencing using the Ligation Sequencing kit (SQK-LSK108; ONT, Oxford, UK) and NEBnext DNA Repair kit reagents (NEB, MA, USA) according to the manufacturer's instructions. The other library was shipped frozen to the University of Oregon (Eugene, OR, USA) and prepared for Illumina Novaseq sequencing using the NEB Ultra II kit (NEB, Ipswich, MA, USA) according to the manufacturer's instructions with a few modifications by the Genomics and Cell Characterization Facility (GC3F) at the University of Oregon (Eugene, OR, USA). DNA was mixed with the fragmentation reagents as described in the kit instructions and

**Table 1** DNA extraction protocol evaluation on commonly used DNA extraction methods in other marine invertebrate studies. Evaluation is based on difficulty, cost, yield, quality, and toxicity.

| Protocol | Difficulty (easy, medium, hard) | Cost per extraction | DNA yield ($\mu g$) | DNA quality (Good=HMW Band, Medium=HMW band with smear, Bad=fragmented smear) | Toxicity (high, medium, low) |
|---|---|---|---|---|---|
| Qiagen Blood and Tissue Kit | Easy | $3.49 | $0.14 \pm 0.084$ | Bad | low |
| Zymo Magbead HMW Kit | Medium | $3.15 | $13.38 \pm 2.06$ | Medium - inconsistent | low |
| PCI Protocol | Hard | $1.83 | $2.84 \pm 1.27$ | Good | high |
| Salting Out Protocol | Medium | $1.50 | $27.78 \pm 9.45$ | Good | low |

fragmented at 37 °C for 8 min, and end repaired as described in the manual. The end repaired sample was mixed with the ligation buffer and enhancer from the NEB kit, and 2.5 µL of 15 µm pre-annealed Tru-Seq style Y-adapter and ligated for 15 min at 20 °C. The sample was cleaned with 2, 0.75× bead cleans to remove adapter dimer. Samples were quantified by qPCR and loaded on the NovaSeq. Protocol methods can be found in the SI 13–17.

## Whole genome sequencing

For Nanopore MinION sequencing, each library was loaded onto an R9 flow cell (FLO-MIN106, ONT). Priming and loading of the SpotON Flow Cells were performed using the standard protocol (sequencing gDNA (SQK-RAD004 or SQK-LSK109) Protocol) (*Lu, Giordano & Ning, 2016*). MinION sequencing was operated with MinKNOW v5.2.13 without basecalling (SI 17). Each flow cell was sequenced until <2 pores were sequencing. Basecalling was conducted on the raw Nanopore MinION reads using Guppy v6.2.1 and high accuracy mode (<5% error rate). Illumina Novaseq sequencing was conducted by GC3F at the University of Oregon using the NEBNext Ultra 2 DNA Library Prep kit for Illumina with slight modifications as stated above. All methods and protocols described above can be found in the Supplemental Information.

## RESULTS

### Extraction performance

Each extraction method was evaluated based on its difficulty, cost, yield, quality, and toxicity. Regarding DNA extraction difficulty (Table 1), the Qiagen kit was considered "easy" due to its lack of complex pipetting steps, while the Zymo kit and Salting out protocol were considered "medium" due to steps such as DNA washing and drying that required proficient lab skills. The PCI protocol was considered "hard" because it required steps with precise pipetting skills such as layered toxic solutions and under a fume hood. Comparing the cost per extraction of each method revealed the PCI and Salting out protocols, which required separate purchasing and assembly of reagents, to
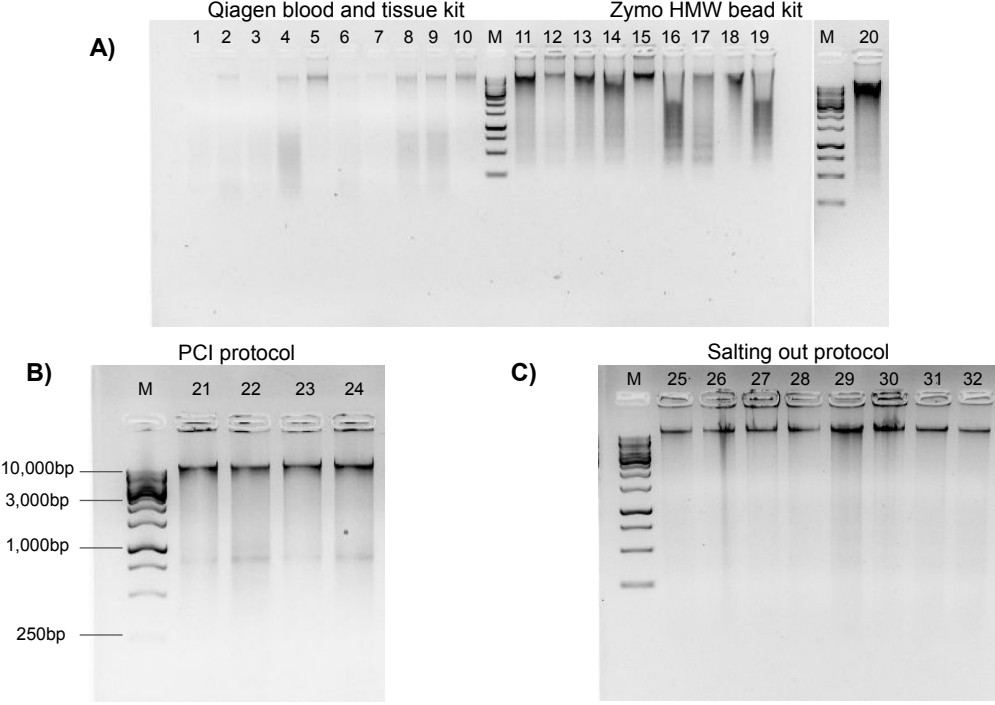

**Figure 2 Genomic DNA extractions conducted on flash frozen Kellet's whelk tissue on 1% gel with ~250 ng gDNA (M = 1 kb DNA ladder).** (A) Wells 1–10 display Qiagen Blood and Tissue kit, wells 11–20 display Zymo Quick-DNA HMW Magbead kit. (B) Wells 21–24 display PCI protocol (~400 bp bands are most likely RNA contaminants due to lack of RNase use in protocol) with 1kb DNA markers (C) Wells 25–32 display modified Salting out protocol.

be approximately half as expensive as the Qiagen and Zymo kit methods, which contain pre-assembled reagents with the kit purchase (Table 1). DNA yield was broadly low in all samples prepared by the Qiagen Blood and Tissue kit (Table 1). DNA extractions produced by the Zymo kit produced optimal DNA yields (>1 µg of DNA) (Table 1). The Salting out protocol produced high and more variable DNA yields with a standard deviation of 9.45 µg (Table 1). The PCI method produced DNA yields lower than either the Zymo kit or salting out protocol, but higher than that by the Qiagen Blood and Tissue kit (Table 1).

The Qiagen and Zymo kits produced poor quality extractions. Samples prepared by the Qiagen Blood and Tissue kit had DNA degradation across most samples as observed by "digested" smears and low molecular weight bands on the agarose gel electrophoresis (Fig. 2). DNA extractions produced by the Zymo kit resulted in inconsistent HMW bands with DNA degradation (Fig. 2). The PCI method produced HMW bands with little DNA degradation (Fig. 2). The Salting out protocol produced consistent HMW bands without DNA degradation across all samples (Fig. 2). All methods had relatively low toxicity, except the PCI method which required the use of phenol and chloroform, which are dangerous chemicals that can be absorbed through the skin and are suspected mutagens. The Salting out protocol was selected as optimal among the four methods, due to its lack of difficulty, lowest cost per extraction, low toxicity, high DNA yield, and best quality HMW DNA.

**Table 2  Salting out protocol validation with 16 samples from different tissue types, location (see Fig. 1), and year of collection.** Resulting DNA yield, DNA quality, and DNA purity based on A260/280 and A260/230 ratios are reported. Average DNA yield = 35.7 ± 27.48μg, A260/280 =1.85 ± 0.046 , A260/230 = 1.81 ± 0.27. See DNA quality descriptions in Table 1.

| Sample number | Tissue type | Site | Year | DNA yield (μg) | DNA quality (Good, Medium, Bad) | DNA purity | |
|---|---|---|---|---|---|---|---|
| | | | | | | 260/280 | 260/230 |
| 1 | Adult | POD | 2015 | 27.71 | Good | 1.89 | 2.13 |
| 2 | Adult | CAK | 2015 | 72.87 | Medium | 1.74 | 1.54 |
| 3 | Adult | PUB | 2015 | 26.33 | Good | 1.86 | 1.89 |
| 4 | Adult | ISM | 2015 | 28.7 | Good | 1.89 | 2.24 |
| 5 | Adult | PAV | 2015 | 107.71 | Medium | 1.91 | 2.23 |
| 6 | Adult | JAL | 2016 | 13.43 | Bad | 1.84 | 1.81 |
| 7 | Adult | MON | 2016 | 13.67 | Medium | 1.85 | 1.81 |
| 8 | Adult | RHR | 2016 | 2.97 | Good | 1.82 | 1.83 |
| 9 | Adult | COJ | 2016 | 67.96 | Bad | 1.9 | 2.15 |
| 10 | Recruit | POD | 2015 | 31.32 | Medium | 1.86 | 1.79 |
| 11 | Recruit | ISG | 2015 | 20.16 | Medium | 1.84 | 1.53 |
| 12 | Recruit | GUI | 2016 | 30.2 | Good | 1.85 | 1.6 |
| 13 | Recruit | BIC | 2016 | 7.4 | Medium | 1.83 | 1.33 |
| 14 | Recruit | MON | 2016 | 40.35 | Medium | 1.87 | 1.87 |
| 15 | Recruit | RII | 2016 | 27.64 | Bad | 1.76 | 1.53 |
| 16 | Recruit | WHC | 2017 | 52.73 | Bad | 1.87 | 1.71 |

## Validation

To test the reliability of the Salting out protocol, it was validated on 16 Kellet's whelk adult and recruit foot tissues from frozen samples collected across the species' entire biogeographic range over multiple years (Table 2; Fig. 1). DNA yield of these extractions was broadly high, with an average of 35.7 ± 27.48 μg (Table 2). HMW bands were evident in all extractions, but signs of digestion/degradation were also evident (Fig. 3). The purity of these DNA extractions were evaluated using the samples absorbance ratios at A260/280 and A260/230. The average A260/280 ratio of these extractions was 1.85 ± 0.046 and A260/230 was 1.81 ± 0.27 indicating pure DNA with, potentially, some ~230 nm absorbing contaminants.

## Scaling

When the Salting out protocol was tested on 3,325 samples, the randomly sampled DNA extractions (252 DNA extractions from all different locations, tissue type, and years) resulted in a range of quality, with most falling into high enough quality for downstream GT-seq and RAD-seq applications. The overall quality of each extraction fell into three groups; Digested (8.7%), HMW + Degradation (29.8%), and HMW (61.5%) (Figs. 4; 5). The majority (81.3%) fell into the groups optimal for RAD-seq and GT-seq; high quality HMW group or HMW + Degradation. The remaining Digested samples (8.7%), would be the least likely to produce sufficient RAD-seq and GT-seq libraries. DNA concentration

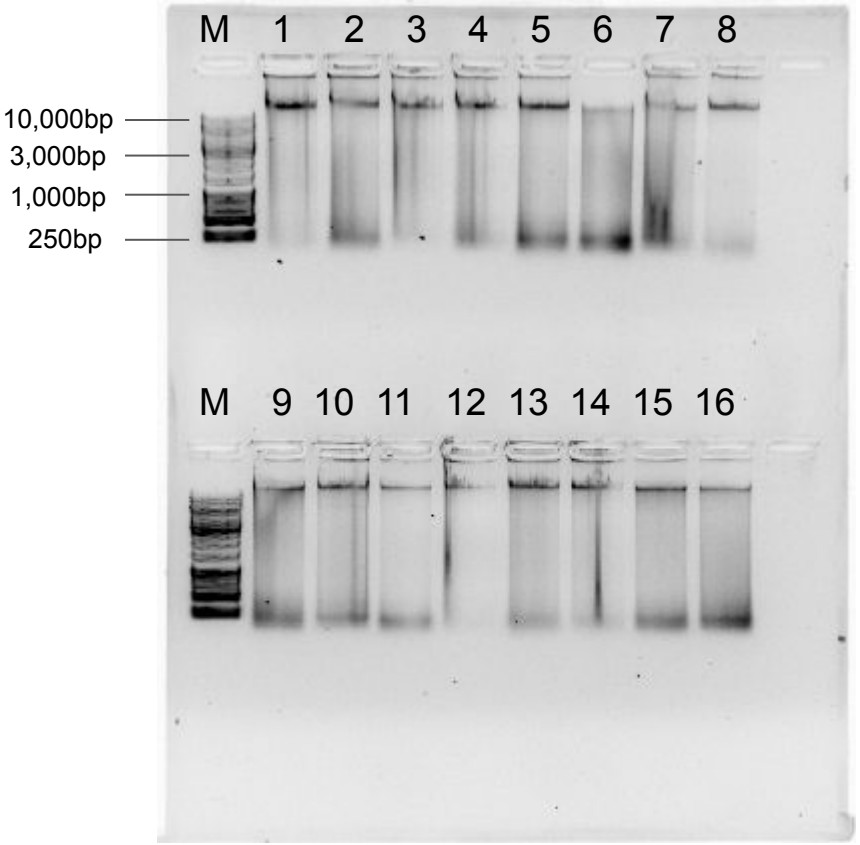

**Figure 3** Genomic DNA extractions using the Salting out protocol on 16 adult foot tissue or recruit samples from frozen samples collected in different years (2015, 2016, and 2017) and locations (POD, CAK, PUB, ISM, PAV, JAL, MON, RHR, COJ, POD, ISG, GUI, BIC, RII, WHC; see sample locations and year in Table 2 and location of site code indicated in Fig. 1) on a 1% gel with ∼250 ng gDNA (M = 1 kb DNA ladder).

was found to be correlated with DNA quality (Fig. 4). This result is most likely due to smaller sample sizes that may have undergone DNA degradation more rapidly during the storing and thawing phase of tissue.

## RAD-seq and GT-seq

A subset of 172 of the scaled-up DNA extractions were analyzed *via* RAD-seq, producing a total of 793,871,366 reads that passed quality assessment. Among the sequenced samples, nine (5.2%) exhibited a total read count lower than 410, with the remainder displaying a notably higher count of over 250,000 reads. All reads were further filtered by the presence of the anticipated RAD-seq cut site, resulting in an average of 78.9% of reads retained after all filtering from the raw reads (Fig. 6). A subset of 96 of the scaled-up DNA extractions were also analyzed with GT-seq. Twenty five (26%) generated >90% GT, 15 (15.6%) generated 30–90% GT, and 56 (58.3%) generated <30% GT (Fig. 6). Approximately half of the samples (49%) had raw read counts below 100,000 reads indicating unsuccessful

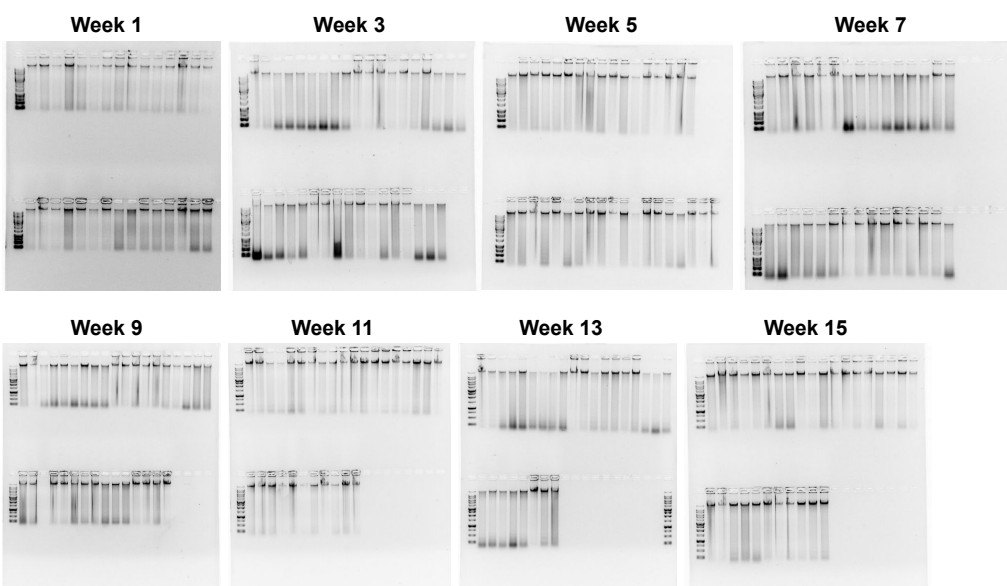

**Figure 4  Quality check gel electrophoresis on randomly selected genomic DNA extractions bi-weekly.** 1% gel with ~250 ng of DNA and 1kb or XL DNA ladder in the first well of each row. See Supplemental Information for sample location, concentration, and quality.

sequencing. In general, the number of raw reads per sample decreased with lower % GT (Fig. 6). No correlation was found between successful genotyping and sample location or DNA yield (Table S6). After the same samples were cleaned using the Zymo Clean and Concentrator there was a 49% increase in successful genotype calling (>90%). Of the samples, 72 (75%) were above 90% GT, 13 (13.5%) were between 30 and 90% GT, and only 11 (11.5%) under 30% GT (Fig. 6). The number of raw reads across each sample was far more consistent with only one sample having less than 100,000 reads, a 50% increase in samples with optimal raw read counts after cleaning (Fig. 6).

## Genome sequencing

The DNA from the Salting out protocol was sequenced on the Nanopore MinION and the Illumina Novaseq. The first library prepared using the rapid sequencing kit resulted in a failed sequencing run (Table 3). The sequencing run produced an estimated 14.01 k reads and 8.94 Mb bases, yet when finalized using the MinKONOW software, no data was produced (Table 3). Both Nanopore MinION libraries prepared using the Genomic DNA Clean Concentrator kit, the PacBio Short Read Eliminator kit, and the ligation sequencing protocol produced successful whole genome sequencing (Table 3). Both sequencing runs produced more than 80 times the estimated number of bases from the rapid sequencing kit run (Table 3). The KK_Ligationseq library produced a total of 2.6 Gb from 446.26 k reads and the KK_Ligationseq2 library produced a total of 1.4 Gb from 98.57 k reads of data that passed high quality base calling using Guppy. Each run had an estimated retention rate of 80.02% and 87.53% for reads that passed high quality filtering, respectively (Table 3).
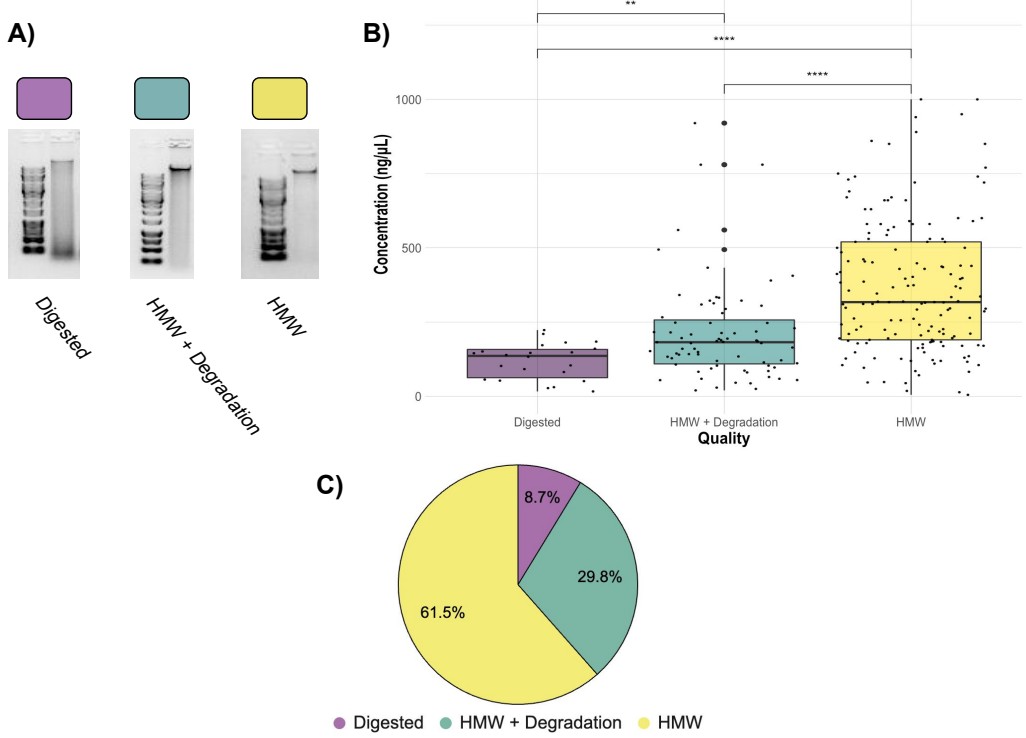

**Figure 5** **Assessment of 252 randomly selected DNA extractions (A) Examples of Digested, HMW + Degradation, and HMW quality genomic DNA extractions determined by gel electrophoresis (B) Quality assessment of genomic DNA extractions.** Average concentration of Digested extractions (22 total samples), HMW + Degradation extractions (75 total samples), and HMW extractions (155 total samples). The top of the box represents the 75th percentile and the bottom represents the 25th percentile. Whiskers represent the highest and lowest values of the sample group. Kruskal Wallis tests are indicated by horizontal lines and asterisks (****: $p \leq 0.0001$, **: $p \leq 0.01$). (C) Pie chart displaying percentage of the 252 randomly-selected samples of the 3,325 DNA extractions within each quality assessment group: Digested, HMW + Degradation or HMW (Table S2).

The DNA from the salting out protocol extraction was also subjected to sequencing on the Illumina Novaseq. The same DNA cleaning and selection method used to produce successful libraries for nanopore sequencing was used. The Illumina Novaseq run produced 715.94 million reads and 113.83 Gb of bases. After filtering, there were 682.18 million reads and 104.85 Gb of bases. The filtered data consisted of 88.78% Q30 bases and 95.27% Q20 bases.

## DISCUSSION

Investigating the spatial and temporal scales of genetic variation among populations of non-model organisms, such as marine invertebrates, can reveal emergent, novel patterns and drivers of population connectivity in a marine system, with significant implications for science and management (*Cowen et al., 2007*; *Fogarty & Botsford, 2007*). Recent advances in genome sequencing technologies hold tremendous potential for genomic research across the tree of life, particularly in under-represented organisms.

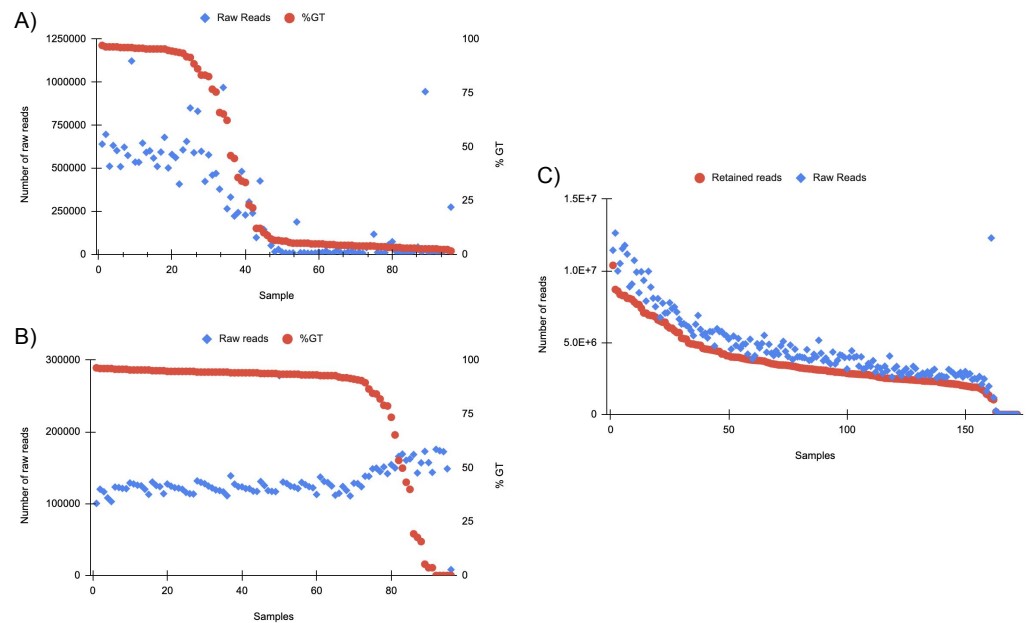

**Figure 6** (A) GT-seq tested samples in order of percent genotyped (%GT) from scaled DNA extractions. 96 total samples were tested (see Supplemental Information). Varied read distribution indicates unsuccessful sequencing especially in samples 40+. (B) GT-seq on the same 96 samples after Zymo clean and concentrator. Samples are ordered by %GT. Distribution of raw reads across samples indicates much more consistent and successful sequencing results. (C) Samples tested on RAD-seq platform. Reads that passed the quality filter and contained the RADcutsite are indicated by Retained reads. Reads are ordered by retained reads. Overall, an average of 78.84% of reads were retained after filtering for quality and RADcutsite. 94.77% of samples produced >180,000 raw reads.

**Table 3** Each sequencing run's output from MinKNOW sequencing software (estimated reads and estimated bases) and Fastp (data produced post-filtering and reads passed filtering). Reads produced by the KK_Ligationseq and KK_Ligationseq2 libraries that passed quality filtering were 80.02% and 87.53% respectively indicating similar quality of sequencing for both libraries. Lower yields for the second round of ONT ligation sequencing library was most likely due to an observed air bubble within the sequencing pore.

| Sequence run | Estimated reads | Estimated bases | Data produced post-filtering | Reads passed filtering |
|---|---|---|---|---|
| KK_Rapidseq | 14.01 k | 8.94 Mb | 0 B | 0 |
| KK_Ligationseq | 557.70 k | 1347 Mb | 2.6 Gb | 446.26 k |
| KK_Ligationseq2 | 112.61 k | 724 Mb | 1.4 Gb | 98.57 k |

While these new technologies hold the capacity for sequencing numerous non-model species. It is essential to recognize that sequencing many organisms, especially marine invertebrates, remains inherently challenging (*Chakraborty, Saha & Neelavar Ananthram, 2020*). Marine organisms retain chemical composites within their tissue that interfere with both common DNA extraction methods as well as many downstream sequencing techniques (*Boughattas et al., 2021*). Moreover, they often exhibit notably high heterozygosity and large numbers of repetitive regions within their genomes (*e.g.*, transposable elements),

underscoring the need for DNA extraction methods capable of producing high molecular weight DNA for long-read sequencing to overcome these genome assembly limitations (*Angthong et al., 2020*). Our objective was to test four common DNA extraction methods using the non-model organism Kellet's whelk, generate an optimal protocol for producing HMW DNA for next generation sequencing, apply this protocol to a large collection of samples, and test its suitability for whole genome sequencing as well as RAD-seq and GT-seq platforms across large sample sizes. Among the four tested DNA extraction methods, the Salting out protocol stood out by producing the highest quality genomic DNA, while also being cost-effective and high yielding (Table 1). The Salting out protocol was improved by the use of simple low agitation mechanical tissue homogenization, a set duration for chemical tissue lysis with proteinase K, and an overall lack of DNA disruption by minimizing the use of pipette mixing, vortexing, and filter columns (SI 10). The necessity of these improvements became evident during initial testing, as they played a vital role in producing HMW DNA, given that different methods for homogenization and tissue lysis yielded varying outcomes (Fig. 2). It is unclear why the Salting Out method was most successful in extracting consistent HMW DNA; it is possible that secondary metabolites specific to Kellet's whelk may interfere with the efficacy of these extraction methods/kits differently. Importantly, we validated this method using samples collected 3–5 years prior to extractions and stored at −80 °C, which resulted in genomic DNA with high purity, as evident by A260/280 and A260/230 ratios suitable for population genetic platforms (Table 2). Notably, any signs of degradation observed in these samples collected years ago from disparate locations (Fig. 3) were not evident in samples produced in the lab from fresh tissue (Fig. 2). This finding strongly suggests that compromised DNA quality stems not from the extraction method but rather from DNA degradation during tissue collection or storage. Conducting extractions on all 3,325 tissue samples using the Salting out protocol produced DNA with sufficient quality for RAD-seq and GT-seq, despite some variability among samples (Fig. 4). We attribute this variability to potential factors such as human error, low tissue input, and suboptimal storage conditions (frozen on dry ice and subjected to multiple transfers before being stored at −80 °C). Nevertheless, the Salting out method yielded a majority (81.3%) of DNA extractions of optimal quality for RAD-seq and GT-seq platforms. We interpreted the quality of the DNA as measures of the potential success in RAD-seq and GT-seq platforms (81.3%) by requiring HMW DNA be present (as shown in gel electrophoresis Fig. 4) and this was reflected in the overall results of the subset testing on GT-seq (75% successful) and RAD-seq (78.9% successful) (Fig. 6). We anticipate that further improvements can be achieved by collecting samples under more consistent conditions and employing robotics rather than manual operators (trained students, see Methods), which may elevate the overall DNA quality across samples. While the Salting out protocol produced HMW DNA extraction, it likely did not remove all contaminants, as indicated by the failed ONT rapid sequencing library and initial suboptimal GT-seq libraries (Table 3, Fig. 6). These contaminants may have potentially interfered with the polymerase activity of the MinION Nanopore technology and enzyme activity during library preparation for the Illumina technologies (*Healey et al., 2014*). With the introduction of additional cleanup steps, the ONT ligation sequencing libraries and Illumina Novaseq

library successfully achieved high-quality genome sequencing. Moreover, following the same cleanup process, 80% of the failed GT-seq samples from the initial run produced successful genotyping (>90% call rate) (Fig. 6). The bead cleaning steps in combination with the PCR step employed in the library preparation of the RAD-seq libraries appeared to exhibit resistance to inhibition by co-precipitants during sequencing. This observation implies a cleanup step is necessary before conducting PCR library amplification in order to facilitate successful sequencing.

The issue of inhibition or blockage of Nanopore pores was also observed in the black tiger prawn, *Penaeus monodon*, and believed to be caused by repetitive regions in the genome (*Van Quyen et al., 2020*). Successful Nanopore sequencing occurred after PCR-based library preparation, suggesting that there may be unknown contaminants associated with mollusc DNA that are removed or destroyed during PCR (*Adema, 2021*). Similarly, we encountered comparable challenges in producing successful genome sequencing using the Nanopore MinION without added cleanup and purification. Our cleanup, which did not include PCR steps and maintained the integrity of our HMW DNA, allowing secondary structures to form prior to sequencing, is believed to remove the unidentified contaminant responsible for the failed Nanopore MinION sequencing runs (SI 14–15). Our comparison of DNA extraction methods could have benefited from a method for isolating or quantifying co-precipitants. These co-precipitants, most likely a mucopolysaccharide (*Sokolov, 2000*), were not significantly implicated by the A260/280 ratio (A260/280 = 1.85 ± 0.046) or the A260/230 ratio (A260/230 = 1.81 ± 0.27) and were only clearly evident after sequencing, not being implicated by PCR failure (*Popa, Murariu & Popa, 2007*). To prevent future failed sequencing runs, we recommend implementing a method for detecting these co-precipitants (2D gel-electrophoresis) or ensuring their removal (using the added clean-up, HMW selection kits, or PCR). Other methods known for removing polysaccharides, such as the CTAB method (*Chakraborty, Saha & Neelavar Ananthram, 2020*; *Huelsken, Schreiber & Hollmann, 2011*), should also be explored, as well as other modified methods being published regularly, given the increasing number of non-model marine invertebrates under genetic analysis (*Ardura et al., 2017*; *Panova et al., 2016*; *Angthong et al., 2020*).

By applying whole genome sequencing using GT-seq and RAD-seq to the extensive collection of Kellet's whelk adult and recruit samples from populations spanning the US and Mexico coasts, southern and central California, and the species' historical and expanded range, over multiple years covering ENSO and non-ENSO oceanographic conditions, researchers can hope to increase understanding of population connectivity in this and other coastal marine species with extensive larval dispersal (*Christie et al., 2017*). However, accomplishing such large-scale sequencing data on a non-model marine invertebrate species of this scale is challenging. This study serves as an important contribution towards enhancing our understanding of the performance and potential use of DNA extraction methods for future large-scale genomics.

## ACKNOWLEDGEMENTS

The undergraduate students that conducted the DNA extractions in the Center for Biotechnology at California Polytechnic State University: Jaden Hansen, Anabel Sanchez, Gabriella Richardson, Hanna Jaynes, Tyler Weipert, Alyssa Queen, Kathryn Hutchinson, Olivia Watt, Jordan Reichhardt, Chanel De Smet, and Alli Clark. The RAD-sequencing was carried out by the DNA Technologies and Expression Analysis Core at the UC Davis Genome Center. The GT-sequencing testing was carried out by GTseek with special thanks to Nathan Campbell. The Illumina Novaseq Genome sequencing was carried out by the Genomics & Cell Characterization Facility (GC3F) at the University of Oregon with special thanks to Jeff Bishop.

### Funding

This material is based upon work supported by the National Science Foundation under Grant No. 1924537. The funders had no role in study design, data collection and analysis, decision to publish, or preparation of the manuscript.

### Grant Disclosures

The following grant information was disclosed by the authors:
The National Science Foundation: 1924537.

### Competing Interests

Robert J. Toonen is an Academic Editor for PeerJ.

### Author Contributions

- Benjamin N. Daniels conceived and designed the experiments, performed the experiments, analyzed the data, prepared figures and/or tables, authored or reviewed drafts of the article, and approved the final draft.
- Jenna Nurge performed the experiments, authored or reviewed drafts of the article, and approved the final draft.
- Olivia Sleeper performed the experiments, authored or reviewed drafts of the article, and approved the final draft.
- Andy Lee conceived and designed the experiments, performed the experiments, analyzed the data, authored or reviewed drafts of the article, and approved the final draft.
- Cataixa López conceived and designed the experiments, performed the experiments, analyzed the data, authored or reviewed drafts of the article, and approved the final draft.
- Mark R. Christie conceived and designed the experiments, authored or reviewed drafts of the article, and approved the final draft.
- Robert J. Toonen conceived and designed the experiments, authored or reviewed drafts of the article, and approved the final draft.

- Crow White conceived and designed the experiments, authored or reviewed drafts of the article, and approved the final draft.
- Jean M. Davidson conceived and designed the experiments, authored or reviewed drafts of the article, and approved the final draft.

### Field Study Permissions

The following information was supplied relating to field study approvals (i.e., approving body and any reference numbers):

Field collections were approved by the CDFW under the Scientific Collection Permit 8018.

### DNA Deposition

The following information was supplied regarding the deposition of DNA sequences:

The Nanopore MinION and Illumina NovaSeq data is available at the Sequence Read Archive (SRA): SRR25442105, SRR25442104 (Nanopore MinION); SRR25442103 (Illumina NovaSeq), respectively. The data is also available at NCBI BioProject: PRJNA999368.

### Data Availability

The code described in the Methods is available at Github:

- https://github.com/bndaniel/Genomic-DNA-Extractions-for-Kelletia-Kelletii

- Daniels, B. (2023). Kellet's whelk genomic extractions. Zenodo. https://doi.org/10.5281/zenodo.10058845.

The commands for quality checking were adopted from FastQC (https://www.bioinformatics.babraham.ac.uk/projects/fastqc/) and base calling from Oxford Nanopore Technologies (https://community.nanoporetech.com/downloads).

The R scripts were developed using ggplot2 (https://ggplot2.tidyverse.org/).

### Supplemental Information

Supplemental information for this article can be found online at http://dx.doi.org/10.7717/peerj.16510#supplemental-information.

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
