# Peer review of "Genomic DNA extraction optimization and validation for genome sequencing using the marine gastropod Kellet’s whelk"

_PeerJ, doi:10.7717/peerj.16510_

## Round 0.1 · original submission · Minor Revisions

I agree with the authors that the manuscript deals with an often-overlooked criterion of NGS - genomic DNA extraction. Although the overall writeup is solid, I share similar concerns with the reviewers, i.e., the evaluation strategy of the authors on the four DNA extraction methods/kits based solely on gel band seems insufficient. Kindly provide stronger justification to this.

Reviewer 1 ·

Basic reporting

As a researcher that routinely deals with next generation sequencing of marine crustaceans and other marine invertebrates, I utterly agree with the authors on the challenges in extracting and purifying their genomic DNA. I applaud the authors for developing an optimized protocol that is useful for, at least, gastropods. The scale and length they used to validate the effectiveness of the optimized protocol – up to RAD-seq, GT-Seq and whole genome sequencing. The overall language of the manuscript is easy to read and straightforward. However, I was anticipating to read on the validation of the comparison of the four extraction protocols/kits up to NGS scale, not just one. By doing so, the authors could further validate and show the weaknesses and strengths of each protocol.

Experimental design

The experimental framework is solid, with sufficient replications and in-depth analyses. However, why didn’t the authors check the DNA quality and purity during the comparison stage? Maybe some protocols would produce high quality DNA too. Assessing only via gel bands is insufficient to justify and quantify DNA quality. The authors also did not highlight what are the baseline of DNA quality that can be used for subsequent next generation sequencing.

Validity of the findings

It is quite surprising that Zymo kit produced low quality extractions. I have used Zymo DNA extraction kit and also Zymo RNA extraction kit on crustacean muscles, and obtained clear bands with high yield. Since the authors only tested on whelk tissues, do you think there could be some difference in concentrations of unknown secondary metabolites that are species-specific that could interfere with this? I think this aspect should be included in the discussion as well.
Also, to support the Salting Out protocol, I would suggest the authors to discuss on the methods or ingredients of this protocol that enable it to stand out against the other three compared protocols/kits.

Additional comments

Please double check that all species names are italic. (e.g. line 408).

Reviewer 2 ·

Basic reporting

In this manuscript, the authors developing and testing DNA extraction protocols for Kellet’s whelk (Kelletia kelletii), a subtidal gastropod with ecological and commercial importance, by comparing four DNA extraction methods commonly used in marine invertebrate studies, and the findings may offer a robust and versatile DNA extraction and clean-up protocol for supporting genomic research efforts on non-model marine organisms, to help mediate the under-representation of invertebrates in genomic studies. The manuscript is well-organized and of significant importance, only a few issues need to be addressed for a better clarification.

Experimental design

no comment

Validity of the findings

no comment

Annotated reviews are not available for download in order to protect the identity of reviewers who chose to remain anonymous.

---

## Round 0.2 · accepted · Accept

Thank you for addressing all comments and concerns from the previous review. The manuscript is ready for publication.